# *Djck1α* Is Required for Proper Regeneration and Maintenance of the Medial Tissues in Planarians

**DOI:** 10.3390/cells12030473

**Published:** 2023-02-01

**Authors:** Yongding Huang, Yujia Sun, Yajun Guo, Mengwen Ma, Shoutao Zhang, Qingnan Tian

**Affiliations:** 1School of Life Sciences, Zhengzhou University, Zhengzhou 450001, China; 2Longhu Laboratory of Advanced Immunology, Zhengzhou 450000, China

**Keywords:** *Djck1α*, *Djslit*, mediolateral axis, stem cells, regeneration, planarian

## Abstract

CK1α (Casein kinase 1α) is a member of the casein kinase 1(CK1) family that is involved in diverse cellular processes, but its functions remain unclear in stem cell development. Freshwater planarians are capable of whole-body regeneration, making it a classic model for the study of regeneration, tissue homeostasis, and polarity in vivo. To investigate the roles of CK1α in regeneration and homeostasis progress, we characterize a homolog of CK1α from planarian *Dugesia japonica*. We find that *Djck1α*, which shows an enriched expression pattern in the nascent tissues, is widely expressed especially in the medial regions of planarians. Knockdown of CK1α by RNAi presents a thicker body due to dorsal hyperplasia, along with defects in the medial tissues including nerve proliferation, missing epidermis, intestine disturbance, and hyper-proliferation during the progression of regeneration and homeostasis. Moreover, we find that the *ck1α* RNAi animals exhibit expansion of the midline marker *slit*. The eye deficiency induced by *slit* RNAi can be rescued by *ck1α* and *slit* double RNAi. These results suggest that *ck1α* is required for the medial tissue regeneration and maintenance in planarian *Dugesia japonica* by regulating the expression of *slit*, which helps to further investigate the regulation of planarian mediolateral axis.

## 1. Introduction

Substrate phosphorylation, mediated by protein kinases, is a common modification that plays an essential role in the regulation of various cellular functions [1,2]. The casein kinase 1(CK1) family is a group of evolutionarily conserved monomeric serine (Ser)/threonine (Thr) protein kinases found in eukaryote organisms ranging from yeast, plants, algae, and protozoa to mammals [3,4,5,6,7]. CK1 participates in numerous developmental signaling including the Hedgehog (Hh) and Wnt signaling pathways [8,9]. It has been implicated in a variety of cellular functions, including DNA processing and repair, cell division, cytoskeleton dynamics, membrane receptor trafficking, circadian rhythms, and cell differentiation [10,11,12,13]. Seven isoforms from the CK1 family are found in mammals (α, β, γ1, γ2, γ3, δ, and ε), whereas eight are found in *Drosophila*, and 87 members are found in *C. elegans* [14,15,16]. All members have a highly conserved kinase domain in mammals [17]. Among them, Casein kinase 1α (CK1α) has been confirmed to be crucial for the phosphorylation of β-catenin at the destruction complex (a known major component of the Wnt pathway) [4,18,19]. However, the function of CK1α in stem cells was not yet defined. Data from several previous studies suggest that CK1α participates in polarity development, especially the regulation of axial polarity whereas understanding its regulation remains challenging [20,21,22].

Planarian is a member of the phylum Platyhelminthes with an almost unparalleled ability to regenerate the missing tissues and rebuild the whole body after being injured. It is considered to be an outstanding model for studying regeneration and polarity reconstruction [23,24,25,26]. As constitutive flat animals, planarians have sophisticated anatomy, including the brain, eyes, intestine, musculature, and epidermis, all arranged in complex and appropriate patterns [27,28]. It can regenerate a complete individual from almost any irregularly shaped fragment, owing to an abundant adult somatic stem cell population (called neoblasts, approximately ten percent of all animal cells) [25,29,30,31,32]. Furthermore, a process known as tissue turnover means that the neoblasts of adult planarians constantly produce new cells to replenish dying ones, demonstrating the remarkable capability to regenerate in planarians [25]. The tissue turnover and regeneration process require the new cells to differentiate and to form in the location in the body correctly, for example, new tail cells need to form in the posterior pole, and new eye cells need to form in the heads [33]. Position control genes (PCGs), some of which are involved in tissue regeneration and polarity reconstruction, are required for the orderly expression of several genes throughout the dynamic and complicated tissue turnover and regeneration process [34,35,36]. PCGs control tissue identity along the anterior–posterior (AP) axis, the dorsal–ventral (DV) axis, and the medial–lateral (ML) axis in planarians. For example, a canonical β-catenin-dependent Wnt signaling pathway is required for anteroposterior blastema polarity in planarian regeneration [37,38,39]. Upregulation of the canonical Wnt negative regulators *notum* and *APC* causes the regeneration of ectopic tails whereas downregulation of Wnt pathway components *β-catenin-1, Evi/wntless, wnt1, Dvl-1/2* or *teashirt* causes the ectopic heads [40,41,42,43,44]. The bone morphogenetic protein (BMP) pathway has been shown to play an important role in the dorsoventral axis. After BMP pathway silencing, the DV axis in planarians is disrupted [45,46]. The planarian midline is regulated by *slit*, which acts as a repulsive cue required for proper midline formation. The suppression of *slit* can cause the medial tissue defects and a collapse of regenerating central nervous system (CNS) [47]. It is now well established that planarian presents a robust system for studying polarity; however, the mechanisms of controlling positional information have remained unclear.

In our work, we characterize a homolog of CK1α, a member of the casein kinase 1 family, from planarian *Dugesia japonica*. We show that RNA interference targeting *ck1α* causes a wide range of regeneration and tissue homeostasis defects such as intestine disorder, epidermis absence, and neurologic abnormality in medial tissues. Meanwhile, the division of phospho-H3 mitotic cells is increased in both regenerating and intact *ck1α* RNAi planarians. CK1α RNAi in planarians revealed some phenotypes that were opposite to the *slit* RNAi phenotypes. We propose that *ck1α*, which regulates the expression levels of the *slit*, is required for the regeneration and maintenance of medial tissues in planarians.

## 2. Materials and Methods

### 2.1. Planarian Culture

An asexual strain of planarian *Dugesia japonica* was cultured in autoclaved stream water at 22 °C, as previously described [48,49]. Planarians 5–9 mm in length were selected and starved for at least 7 days before the experiments.

### 2.2. Gene Identification and Cloning

Ck1α sequences from planarian *Dugesia japonica* species were identified in the planarian transcriptome [50]. A pair of specific primers (*Djck1αF* and *Djck1αR*) were designed to amplify the *Djck1α* from cDNA (extracted from intact planarians). The *Djck1α* sequence was cloned into PMD-19-Vector (Takara, Kyoto, Japan) for further experiments. All primers used for *Djck1α* and *Djslit* cloning and dsRNA generation are listed in Appendix A.

### 2.3. RNAi Experiments

The double-stranded RNAs of *Djck1α* were synthesized by in vitro transcription as previously described [49,51]. The dsRNA was dissolved in RNase-free H_2_O (water treated by DEPC). The *Djck1α* dsRNA was injected into the experimental planarians using a Drummond microinjector, while the control animals were injected with water treated by DEPC. The dsRNA concentration used for injection was 2 µg/µL. The volume of dsRNA was 0.8 µL, once per planaria. The injection site was mainly on the dorsal side, which varied day to day, and the injection rate was once each day. Animals were injected consistently for a week, unless otherwise stated, and heads and tails were amputated 24 h after the last injection.

### 2.4. Whole-Mount Immunostaining

Whole-mount immunostaining was performed as previously described [52]. In brief, the animals were killed with 5% NAC in phosphate-buffered saline (PBS) for 6 min at room temperature and washed four times with PBS containing 0.1% TritonX-100 (PBST). Then, the animals were fixed in PBST containing 4% paraformaldehyde. Next, the animals were blocked with 10% goat serum in PBST for 2 to 4 h at 4 °C and incubated with primary anti-H3P (1:500; Millipore, Burlington, MA, USA, 05-817R) or anti-synapsin (1:100–1:500 Developmental Studies Hybridoma Bank) antibodies overnight. The secondary antibodies include goat anti-rabbit Alexa Fluor 568 (1:500; Invitrogen, Waltham, MA, USA, 11036) for anti-H3P, and goat anti-mouse Alexa Fluor 488 (1:500; Invitrogen, 673781) for anti-synapsin. Digital pictures were collected using NIS element software (version 4.2.0, Nikon, Tokyo, Japan).

### 2.5. In Situ Hybridization

As previously described, Whole-mount in situ hybridizations (WISHs) were performed with digoxigenin-labeled probes [53]. All samples were hybridized with a DIG-labeled probe at 56 °C for at least 16 h. Colorimetry (NBT/BCIP) was subsequently used to detect the signal.

### 2.6. Quantitative RT-qPCR

Total RNA was extracted using Trizol reagent (TaKaRa, Gunma, Japan), and cDNA was synthesized from 1 μg of total RNA with oligo-dT primers and reverse transcriptase based on the manufacturer’s protocol (TaKaRa). Quantitative real-time PCR was performed as previously described [53]. Three replicates were run in parallel for each condition. Data were normalized to expression level elongation factor 2 (*Djef2*) [54]. The 2^−∆∆CT^ method, which was described by Schmittgen and Livak, was used to calculate expression ratios [55]. The primers used for quantitative real-time PCR are listed in Appendix A.

### 2.7. Statistical Analysis

The data for gene expression are presented as means ± SD (mean and standard deviation). Statistical analyses were performed using unpaired Student’s *t*-test. One-way analysis of variance was used for analyzing two or more groups of data. Differences were considered significant at *p* < 0.05 level and extremely significant at *p* < 0.01 level.

## 3. Results

### 3.1. Djck1α Expresses in the Middle and Is Required for Normal Tissue Regeneration and Maintenance

To better investigate the role of CK1α in planarian regeneration, we identified a *ck1α* homolog in *Dugesia japonica* (*Djck1α*) and the full-length cDNA of *Djck1α* was obtained by PCR based on the transcriptome (Appendix A) [50]. Then, we performed whole-mount in situ hybridization in regenerative and intact animals to investigate spatiotemporal expression patterns. The animals were amputated before and after the pharynx into three sections: head, trunk, and tail, for regenerating, and were fixed at 1, 3, 5, and 7 days after amputation. The WISH results revealed that the expression of *Djck1α* was detected throughout the body except the pharynx in intact animals, with positive signals found clearly in the midline (Figure 1A). In regenerating animals, a higher expression level of *Djck1α* was detected in the wound region after amputation, and this pattern of expression persisted throughout the regeneration process. By 7 days post amputation, the newly regenerated heads and tails were completed, and *Djck1α* was mainly distributed in the new body parts.

The endogenous *Djck1α* was knocked down by RNAi, and the downregulation efficiency was demonstrated using qRT-PCR (Figure 1B). The RNAi planarians’ regeneration rate was perceived to be slower than that of the controls during regenerating (29/29). The *Djck1α* RNAi planarians were unable to regenerate heads and tails that were the proper size and form. The new heads lacked a clear ‘triangular structure’, and the new tails were shorter (Figure 1C). Dramatically, there was the probability of observing a ‘protruding outgrowth’ in the 21 days regenerative animals below the newly generated head on the dorsal side (8/21) (Figure 1C). In intact animals, there was no obvious difference in appearance in the first week after RNAi; however, a visible ‘white line’ was observed on the dorsal side in the middle of the body starting from the tail at about 14 days after RNAi. As RNAi proceeded, the ‘white line’, which is clearly visible as an ‘outgrowth line’ on the dorsal side, lengthened and became more apparent from the tail to the head (Figure 1D). It is possible that *Djck1α* plays a substantial role in the regeneration and maintenance of medial tissues based on the unique yet robust expression of *ck1α* and the phenotype induced by silencing *Djck1α* in intact and regenerated animals.

### 3.2. Djck1α Inhibition Causes Abnormalities in Nervous, Intestine, and Epidermis Systems during Regeneration

Since the RNAi of *Djck1α* resulted in the formation of smaller heads and tails with evident outgrowth on their dorsal side, we performed the WISH with *Djpc-2* and the whole mount immunofluorescence for anti-synapsin (Syn) to determine the development of the nervous systems [56,57,58,59].

The RNAi phenotypes observed in head and tail fragments could be detected in the trunk fragments simultaneously. Therefore, except as otherwise noted, the subsequent studies were mainly conducted using the trunks. The results showed that the complete and closed ventral nervous cords (VNCs) existed in the *Djck1α* RNAi animals after 10 days of regeneration, whereas both the width of the VNCs and the distance between two ventral nerve cords increased (Figure 2A). Additionally, we detected that *Djck1α* RNAi animals exhibited more diffuse staining of the cephalic ganglion areas with unclear boundaries in all cases (100%) (Figure 2A).

By day 8, when the control animals had regenerated completely, the *Djck1α* RNAi animals were observed to lose the ingestion behavior. Therefore, we performed whole-mount in situ hybridization to identify the variation of the gut by using probes for *Djporcn-1* [44]. The gut of the control animals consisted of a single branch anteriorly, which bifurcates into two detached posterior branches at the pharynx (Figure 2B). However, in *Djck1α* RNAi animals, intestinal morphology showed more robust expression, and more secondary and tertiary branches throughout the intestinal system (Figure 2B).

To investigate the ‘middle line’ on the dorsal of the intact *Djck1α* RNAi animals, we then performed WISH with *Djvim* (the differentiated mature epidermal cells marker) and *Djlaminb* (the epidermal boundary marker) 10 days after RNAi [60,61,62,63]. Considering the previously observed phenotypes, the planarian epidermis was possibly separated by the ‘medial line’; in the meantime, the WISH results suggested the same possibility. Compared to the control animals, which possess the intact epidermis, the expression of *Djvim* in the *Djck1α* RNAi animals showed conspicuous missing at the ‘medial line’ region (Figure 2C). The expression patterns of *Djlaminb* were consistent with the controls overall; however, some animals (7/12) showed positive signal out of border position (Figure 2C). The WISH results of *Djmhc-a* (a marker of the pharynx in *Dugesia japonica*) showed that *Djck1α* RNAi had no significant effect on pharynx regeneration (Appendix A) [64].

Taken together, these findings provide important evidence for the variation of the medial tissues, suggesting that the cell proliferation and fate determination probably differ caused by the knockdown of *Djck1α*.

### 3.3. Djck1α RNAi Results in Hyper-Proliferation

As stated previously, the constitutive animals, *Dugesia japonica*, possess a highly active pool of adult somatic stem cells called neoblasts, some of which are pluripotent and serve as the cellular basis for regeneration. Cell proliferation and differentiation requires the appropriate choices of fate. Inapposite fates can result in hyper- or hypo-proliferative pathological states [65]. Since CK1α plays a vital role in cell division, the unique phenotype of *Djck1α* RNAi animals may indicate the effect on cell proliferation. Whole-mount immunofluorescence for phosphorylated histone H3 (Phospho-H3 which marks cells during the G2/M transition of the cell cycle) was performed to investigate how *Djck1α* RNAi affects cell proliferation. Probes for *Djpiwi-1* (a molecular probe labeled with undifferentiated neoblasts) were used to analyze the effect of *Djck1α* RNAi on neoblasts [66,67].

We counted the number of phosphorylated H3 cells in the head, trunk, and tail fragments, respectively, at 3 days after amputation, but found no significant difference between *Djck1α* RNAi animals and the controls (Figure 3A). However, by day 8, in regenerating animals, a positive increase in the number of phosphorylated H3 cells was observed in *Djck1α* RNAi animals compared to the controls, especially in the region where outgrowth may subsequently arise (Figure 3A). Meanwhile, increased expression of *Djpiwi-1* was detected in the particular region backing onto the newly regenerative heads and tails (Figure 3A), suggesting that downregulation of *ck1α* may increase cell proliferation in particular organizations.

In intact *Djck1α* RNAi animals, we also found that the number of phosphorylated H3 cells increased at 8 days, which was consistent with the results in regenerating *Djck1α* RNAi animals. Meanwhile, the WISH results of *Djpiwi-1* presented the same upward trend. Similar results were obtained by analyzing the expression levels of the same markers by qRT-PCR. These were highly expressed in intact *Djck1α* RNAi animals from medial to lateral (Figure 3B); furthermore, the region with higher expression levels of *Djpiwi-1* usually occupied the same position as the area in which enation appeared (Figure 3B,C). Overall, these results indicate that the downregulation of *Djck1α* can cause hyper-proliferation in both the tissue turnover and regeneration processes.

### 3.4. Inhibition of Djck1α Affects Tissue Turnover and Generates Medial Tissue Dilation in Intact Planarians

We had shown the hyper-proliferation defects in medial tissue were induced by the *Djck1α* RNAi during regenerating progress. To further explore the role of *Djck1α* in the maintenance of medial tissues in homeostasis, we next detected the nervous, intestine, and epidermis systems in the intact (uninjured) *Djck1α* RNAi animals.

The VNCs showed the same variation as previously described, including the wider distance between two nervous cords and stronger expression of the whole nervous system, especially in the cephalic ganglia (CG), which could be a kind of consequence of cell overproliferation (Figure 4A). Meanwhile, we observed drastic changes in gut morphology in intact planarians, including the exceptional connection at two parallel posterior branches in the middle (5/7), more secondary branches, and increased diffuse background before the pharynx (7/7) (Figure 4B). The *Djvim* WISH results showed that missing epidermis in the middle were timed to coincide to the ‘white line’ phenotype in Figure 1D, evolving most noticeably by day 39 (Figure 4C).

The correct and ordered differentiation of neoblasts is essential for the maturation and maintenance of the planarians. The outgrowth in intact animals on the dorsal side had not yet developed into the epidermis (Figure 4C). In consideration of the distinct middle expression of *Djck1α* and the medial outgrowth phenotypes obtained from *Djck1α* RNAi animals, we speculated that there was likely a kind of relationship between *ck1α* and *slit* (a marker of the planarian midline).

We first analyzed the expression of *Djslit*, which is expressed in a medial domain, as previously reported (Figure 4D) [47]. Next, we detected the discrepancy of expression of *Djslit* in intact *Djck1α* RNAi animals. As predicted, 100% of *Djck1α* RNAi animals showed increased expression of *Djslit*, which was denser and expanded in the medial tissues (Figure 4D). Taken together, these results suggested that CK1α can restrict the expansion of the medial tissues, including *slit* expressing cells. For the knockdown of *Djck1α* caused overexpression of *Djslit*, we next assayed whether the *Djslit* RNAi phenotypes can be rescued by double-RNAi experiments simultaneously. The downregulation efficiency of *Djck1α* and *Djslit* in single and double RNAi animals was demonstrated by qRT-PCR (Appendix A). *Djslit* RNAi animals all exhibited a single eye in the newly regenerated head as previously reported (8/8) (Figure 4E) [47]. Interestingly, *Djck1*α and *Djslit* double RNAi animals (13/22) showed two normal eyes, while the remaining 9/22 *slit/ck1α* RNAi animals showed one central eye (Figure 4E). These data demonstrate that *ck1α* has a restriction on the expansion of the medial tissues by restricting the expression of *slit* in planarians. Then, we performed qRT-PCR to investigate the expression of other PCGs as *wnt5*, *wnt1*, *bmp4* and *β-catenin* after *ck1α* RNAi (Appendix A). We found that with the downregulation of the endogenous *ck1α*, the *slit* gene showed higher expression, and the *wnt1* and *wnt5* genes showed no significant changes, while the expression levels of *β-catenin* and *bmp4* decreased at 8 days post RNAi. We next performed double RNAis with *β-catenin* and *bmp4,* which correspond to AP polarity and DV polarity, respectively (Appendix A). By day 14, *β-catenin* RNAi animals exhibited two heads (5/6), whereas *ck1α*&*β-catenin* RNAi animals showed outgrowths on both heads (3/6) (Appendix A), suggesting that the extra outgrowth caused by *Djck1α* RNAi at the midline is independent of AP polarity. The *ck1α*&*bmp4* RNAi animals showed a more obvious bulge and a thicker body than *bmp4* RNAi or *ck1α* RNAi (Appendix A), suggesting that *ck1α* RNAi enhances the defects related to DV patterning.

## 4. Discussion

Casein kinase 1α (CK1α), a conserved protein that exists and functions in a variety of signaling pathways, has been identified as a therapeutic target in some cancers [68,69]. In *zebrafish*, the expression pattern of *ck1α* is ubiquitously expressed in early stages of the development. At 24 and 48 hpf, cross and longitudinal sections of the embryo show that *ck1α* expression is not uniform and seems to be concentrated in some cell clusters in the brain and neural tube [70]. In *C. elegans,* KIN-19/CK1α has been shown to regulate seam stem cell asymmetric division via a Wnt/β-catenin pathway, as well as seam stem cell terminal differentiation in tandem with the heterochronic/temporal identity pathway [21]. Meanwhile, previous studies have suggested that CK1α has a significant effect on axis polarity. However, in the fields of regeneration, little is known about the role of CK1α.

Planarians have constituted an excellent model in which to study polarity specification. The WNT/β-catenin signaling pathway has been shown to participate in tissue regeneration and polarity re-establishment. β-catenin functions as a molecular switch to decide and maintain anteroposterior identity in the planarian *Schmidtea mediterranea* [40]. Recently, the NR4A or SRC was proved to act on WNT signaling to pattern the planarian anterior–posterior axis [71,72]. The BMP pathway is essential for re-specification and maintenance of the dorsal–ventral (DV) axis. The SLIT protein has been emphasized to regulate the planarian midline in medial–lateral (ML) axis [47,73].

Herein we identified a *ck1α* gene from the planarian *Dugesia japonica* based on the similarity of its predicted product to CK1α proteins from other organisms. RNAi silencing of CK1α resulted in defects in the medial tissues including the brain, ventral nerve cord cells bodies, dorsal epidermis and intestinal tissues, suggesting that CK1α is required for organizing the mediolateral axis (Figure 2A–C and Figure 4A–D). We propose that the defects are a kind of expansion of the medial tissues, for we also detected hyper-proliferation and observed an outgrowth on the dorsal in the middle of the planarian body (Figure 1C,D and Figure 3B,C). Recent studies have implicated that CK1α can enhance the Wnt signaling by interacting with PAWS1 (protein associated with Smad1), a regulator of SMAD1 in BMP signaling that has been shown to induce the formation of a secondary axis in *Xenopus* embryos. These findings indirectly or directly suggest that CK1α plays an important role in polarity establishment during development [22]. Consistent with this, our results suggest that CK1α limits the mediolateral axis into a suitable range, which provides new insights into CK1α in the study of axis polarity in planarians.

Planarian regeneration is guided by molecular mechanisms that restore the identity of the tissue. The mechanisms that control the expression of pole-specific gene programs remain elusive, despite emerging data regarding the expression of signaling molecules for polarity initiation. Abnormalities of the medial tissue defects including the intestine and nervous system resulting from *ck1α* RNAi showed similarities with the consequences of *slit* RNAi (Figure 4A–C). However, the CNS, which was collapsed in *slit* RNAi animals, is labeled with more intensity by *ck1α* RNAi (Figure 4A), indicating that *ck1α* may function as a negative regulator of *slit*. Indeed, an increase in *slit* expressing cells was induced by *ck1α* RNAi (Figure 4D). *Djck1*α and *Djslit* double RNAi can rescue the less eye phenotype caused by *Djslit* RNAi (Figure 4E). Furthermore, the double RNAi phenomena of *ck1α*&*β-catenin* or *ck1α*&*bmp4* show that the *Djck1α* RNAi can cause extra outgrowth when DV (*bmp4* RNAi) polarity is disturbed, suggesting that *ck1α* RNAi can enhance the defects related to DV patterning. Meanwhile, the CK1α protein was previously reported to interact with PAWS1, and this kind of interaction is essential for PAWS1-dependent axis duplication, which suggests that CK1α may be involved in BMP signaling in this manner as well. Therefore, these findings further suggest *ck1α* is required for medial tissues regenerating and maintenance in planarians mainly by regulating BMP signaling. However, the mechanisms by which *ck1α* and *slit* regulate the mediolateral axis and coordinately organize it remains to be determined.

## Figures and Tables

**Figure 1 cells-12-00473-f001:**
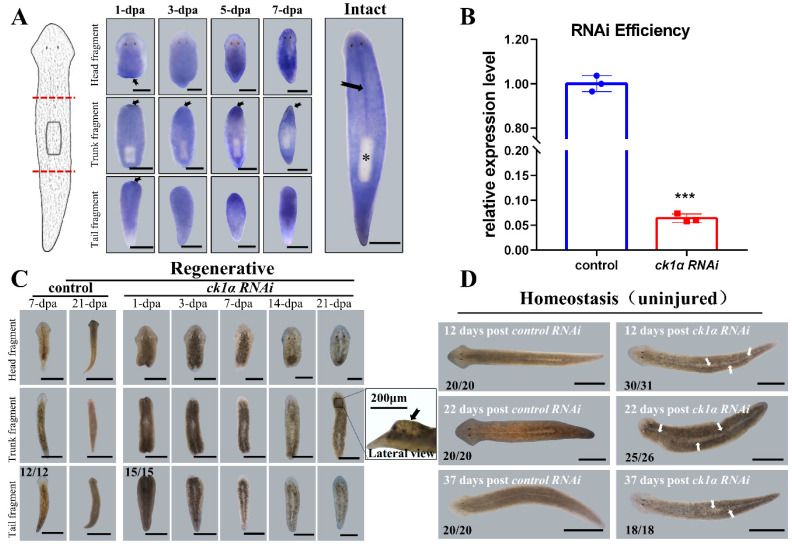
Spatiotemporal expression patterns and RNAi defects of *ck1α* in planarians. (**A**) Spatiotemporal expression patterns of *ck1α* in *Dugesia japonica*. Amputation sites indicated by red dotted lines on the left. Higher expression in the wound regions and the middle, indicated by black arrows. (dpa, days post amputation; *n* = 6 per condition. * pharynx without positive signaling.) (**B**) QRT-PCR for the downregulation efficiency of *ck1α* RNAi at 48 h after the last injection. Unpaired Student’s *t*-test, Mean ± SD. *** *p* < 0.001. Six planarians per group and three replicates per condition. (**C**) Regeneration defects induced by injecting *ck1α* dsRNA into planarians. Animals were amputated pre- and post-pharyngeally after the last injection. Dorsalis protuberances (dotted box and black arrow) in 21 dpa trunks. (**D**) Live images of homeostatic *ck1α* RNAi animals at different times post the last injection. White arrows: visible ‘white lines’ in the middle. Scale bars: 500 μm.

**Figure 2 cells-12-00473-f002:**
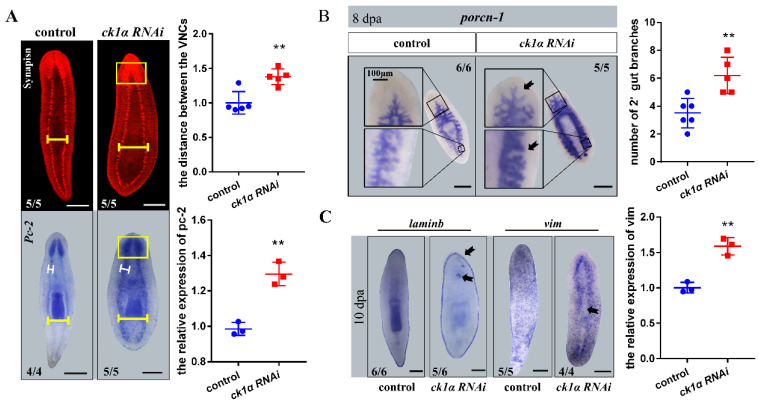
*Djck1α* RNAi animals show drastic differences in the growth of nervous, intestine, and epidermis systems. (**A**) Nervous systems in trunk fragments 15 dpa. Anti-synapsin and *pc-2* riboprobe. *Ck1α* RNAi trunk fragments exhibited blurry cephalic ganglia (Yellow boxes), wider ventral nervous cords (white bar), and a longer distance between them (yellow bar). Upper right: the distance of the VNCs. Bottom right: Quantification of *pc-2*. Unpaired Student’s *t*-test, Mean ± SD. ** *p* < 0.01. (**B**) Intestinal defects (black arrows) in *ck1α* RNAi animals. Stronger expression and more branches of intestines on the right (black boxes). Right: The number of secondary gut branches. Mean ± SD. Unpaired Student’s *t*-test. (**C**) Abnormal expression patterns in trunk fragments (black arrows) of epidermis marker genes (*laminb*, *vim*) by FISH. Right: Quantification of *vim*. Unpaired Student’s *t*-test, Mean ± SD. ** *p* < 0.01. Scale bars: 300 μm.

**Figure 3 cells-12-00473-f003:**
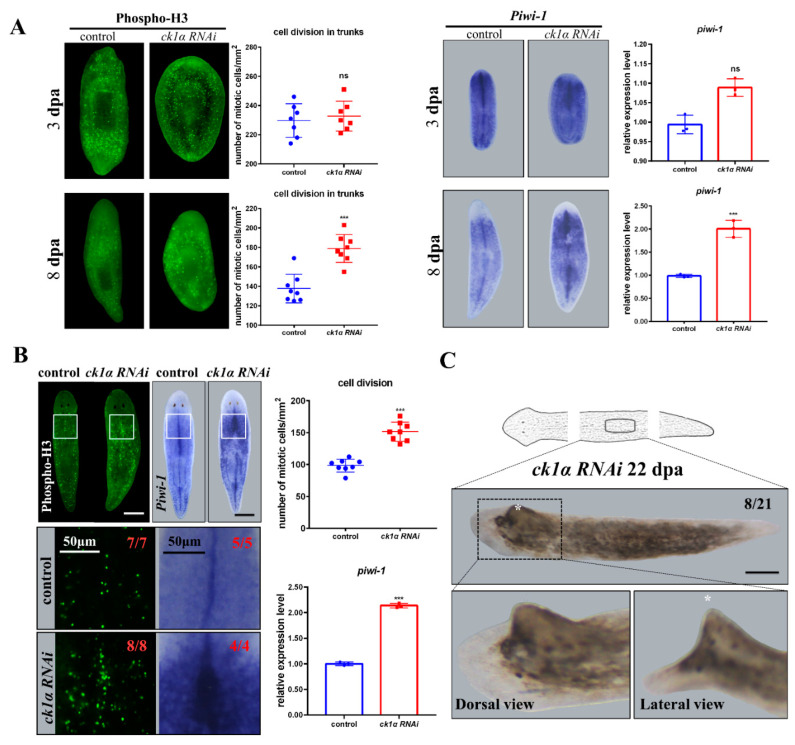
Knockdown of *Djck1α* promotes cell proliferation in both regenerative and intact planarians. (**A**) Phospho-H3 IF and *piwi-1* WISH were performed to detect cell proliferation. RNAi animals were fixed at indicated time points. Left: Quantitative statistical analysis for Phospho-H3 cells. Right: Quantification of *piwi-1*. Unpaired Student’s *t*-test, Mean ± SD. ns *p* > 0.05. *** *p* < 0.001. (**B**) Generation of undifferentiated cells was examined by Phospho-H3 IF and *piwi-1* WISH in intact RNAi animals. Intact RNAi animals were fixed at 8 dpi (days post last injection). The positive cells were quantified in the dorsal region (white boxes). Upper right: Quantitative statistical analysis for Phospho-H3 cells. Bottom right: Quantification of *piwi-1*. Unpaired Student’s *t*-test, Mean ± SD. ns *p* > 0.05. *** *p* < 0.001. (**C**) Regenerating RNAi trunk fragments at 22 dpa. * Outgrowth on the dorsal region. Scale bars: 300 μm.

**Figure 4 cells-12-00473-f004:**
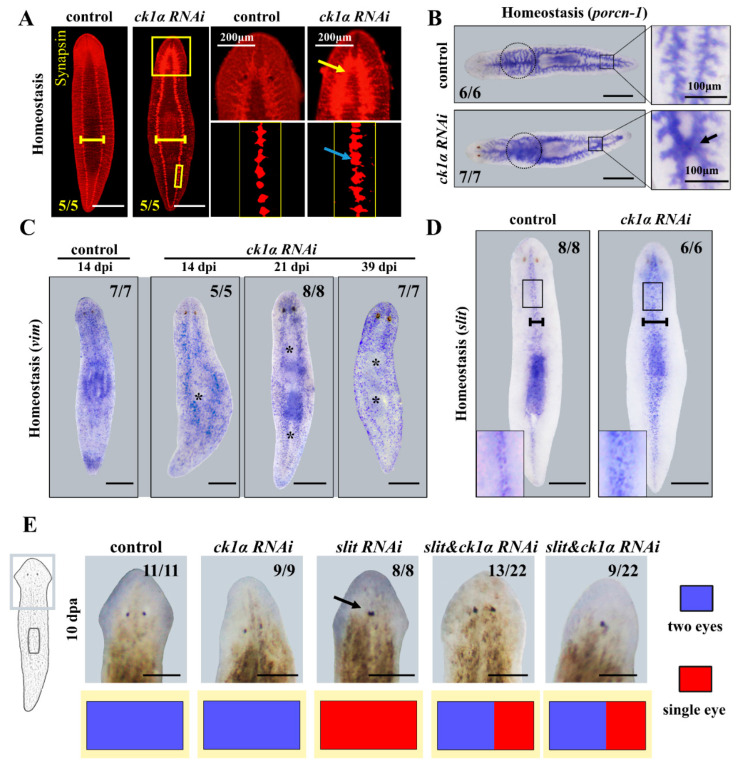
Influence of *Djck1α* RNAi on nerves, intestines, epidermis, and medial tissues in intact planarians. (**A**) Control and *ck1α* RNAi intact planarians at 20 dpi, stained with anti-Synapsin. Yellow arrows point to CG, blue arrows to VNCs. Note the expression and expansion of nervous tissue in *ck1α* RNAi animals (yellow boxes). Yellow bar: distance between VNCs. (**B**) Whole-mount in situ hybridization analysis of *Djporcn-1* mRNA in intact animals. Black arrows indicate the connection in the middle between two posterior branches of the gut. Increased diffuse background was perceived before the pharynx (Black dotted circles). (**C**) Whole-mount in situ hybridization of *vim* as an epidermis marker in intact animals at indicated time points. * Absence of epidermis in the middle of the body. (**D**) Expansion of the midline in intact planarian at 20 dpi. Middle line: *slit* riboprobe. Black bar: the extent of *slit* expression. (**E**) Eye number statistic in regenerating heads after RNAi. Black arrow: single eye in regenerative heads of *slit* RNAi animals. Scale bars: 400 μm; except E, 200 μm.

## Data Availability

Not applicable.

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
