# Peer review of "Djck1α Is Required for Proper Regeneration and Maintenance of the Medial Tissues in Planarians"

_cells, 2023, doi:10.3390/cells12030473_

Round 1

Reviewer 1 Report

In the manuscript “Djck1α Is Required for Proper Regeneration and Maintenance of the Medial Tissues in Planarians” Huang et al. identify the CK1α (Casein kinase 1α) homolog of Dugesia japonica and study their function during planarian regeneration and homeostasis through RNAi. The authors find that ck1α is expressed in several tissues but specially in the midline and new regenerated tissues. RNAi of ck1α produces dorsal overgrowths and defects in several tissues, as the nervous, the digestive and the epidermal. The authors argue that the defects are related with medial structures and conclude that ck1α could be restricting slit expression, since RNAi of both ck1α and slit rescues the slit phenotype.

The study of the role of CK1α in planarians is a topic of interest for the readers of Cells and the study is properly conducted. However, there are several concerns that require further experiments and reinterpretation before the manuscript is ready to be published.

The main concern is about the interpretation of the results obtained. The authors conclude that CK1α has a role in the midline (which they name as medial tissues) but there is no evidence that it has a specific role in the midline, since several defects are also seen in other parts of the animals. For instances, the nervous system is thicker, both brain and VNCs, which suggests that there is a general increase in the number of neurons. The same happens with the gut, also suggesting an increase in the number of cells. From the data presented, CK1α seems to have a very important role mainly during homeostasis, since the phenotype is related to remodelling. Its inhibition produces over proliferation which could be related with the appearance of more cells (neuronal, digestive… it could be quantified) and also a new pharynx (see following points). Whether CK1α has a specific role in the midline region requires further analysis (see following comments).

-        There could be a misinterpretation of the results shown in Fig1 and Fig4. It seems that in CK1α RNAi animals a new pharynx is appearing, as shown in the anti-synapsin, pc-2 and the vimentin staining. This could be a result of the over proliferation seen in these animals. And this could be the explanation of the lack of vimentin staining in this region. If this is the case, it does not mean that the epidermis is disrupted (one of the conclusions of the authors) but that a new pharynx is appearing, and in these regions is normal that vimentin signal is less visible. In Figure 4C the asterisks in vimentin staining in 39 days animals clearly seem to correspond to 2 pharynxes.

A marker of the pharynx must be used to clarify whether a second pharynx is formed in a % of CK1α RNAi animals, because this would change completely the interpretation of the role of CK1α in the midline.

-        In figure 1C the animals are not shown in the same DV position, thus no conclusion can be taken. In the laminb staining the control is D and the RNAi is V (the signal could correspond to a new opening because a new pharynx is appearing?). In the vimentin staining that RNAi could be also V and not D as the control. Furthermore, the pattern of Vimentin is not only different in the midline, as the authors argue, but also in other regions.

-        In Fig 3, the images of H3P show an increase in mitosis everywhere, not only in the midline. And in fact, the mitosis seen in Figure 3C in the upper midline could correspond to the region where a new pharynx will appear. And the posterior midline seems not affected in terms of H3P.

-        In Figure 4- The ‘diffuse background before the pharynx’ (Figure 4B) could correspond to the appearance of a new pharynx.

-        The rescue of the slit phenotype through CK1α inhibition is very interesting. However, the authors need to show that both genes are inhibited at the same level in the single and double RNAis. Furthermore, since there seems to be a problem of over proliferation in general, it could also be that other PCGs as wnt5, wnt1, BMP, etc.. are also overexpressed. This should be investigated. If this is the case, the double RNAis with other PCGs would be also required if the message related with the midline is sustained.

Specific comments:

-        Figure 1 C. Controls of 21 days of regeneration are required.

-        Figure 1 D. Controls of 37 days of homeostasis are required. Furthermore, magnifications of the overgrowths would be also very informative.

-        The scale bar must be revised in all figures. For instance, in Fig 1C, the 21 days head seem to have the same size that the ones at 1-3 days of regeneration, and this is not possible. Also, the controls seem to be smaller, but this could be because in the image a complete regenerated animal is shown, so the scale is different. Please, revise.

-        Quantification of the observations is required: 1) the distance between the VNCs (Fig 2 A), 2) appearance of more secondary and tertiary branches in the gut, 3) In Figure 4- the connection of the two posterior branches; in which % of animals?

Quantification of the number of cells (epidermis, gut, nervous system…) would be extremely useful to understand the phenotype and the role of CK1α in planarians.

-        Figure 3 is a mess. First, the images should be referred in the text following the same order of appearance than in the figure. Second, quantification should be shown next to the corresponding images.

Review interpretation of the data:

-        ‘…blurry brain with unclear boundaries which may correspond to incomplete heads…’ This is not what is observed. The brain appears labelled with more intensity, but there is no evidence of incomplete heads.

-        ‘However, in Djck1a RNAi animals, intestinal morphology showed more robust expression and differentiation, ….’ – Expression yes, but differentiation is something that cannot be evaluated with this marker.

-        ‘However, the CNS, which was collapsed in slit RNAi animals, is expanded by ck1α RNAi (Figure 4A)’- the CNS is not expanded in ck1α RNAi animals, but it seems labelled with more intensity, suggesting that the density of cells is higher. An expansion of the CNS is observed dafter wnt5 inhibition, and this is not the same phenotype observed with ck1α inhibition.

-References on the role of wnt elements in planarians are missing, even more considering the possibility of a pharynx duplication (role of evi-wntless, wnt11-2…).

-The manuscript needs an extensive revision of the English grammar and style. Some paragraphs, as the one in lines 221-228 need to be rewritten.

Reviewer 2 Report

This is a novel, robust, and very well-presented manuscript on the role of CK1a in planarian regeneration and homeostasis. Knockdown of CK1a in planaria disrupts particularly medial tissue regeneration after head/tail amputation, as well as in homeostatic tissue turnover. Expression of CK1a and of other tissue markers is shown clearly. An effect of CK1a on regulating the spatial expression of slit is shown, linking to the phenotypic changes seen in CK1a RNAi planaria.

A clear and detailed background is provided. The methodology and results presented are of excellent quality. The conclusions add significantly to our understanding of the role of CK1a in body axis patterning and regeneration, suggests a connection to slit in these processes, and provides further work to be investigated.

Language

Generally very good language and understandable, with excellent scientific content in the writing, but there are occasional slips in meaning/grammar that need to be addressed for clarity. Some examples in the abstract are given below, but this needs to be addressed very occasionally throughout the manuscript.

Line 9 A clearer sentence would be “but its functions remain…”

Line 10 “whole body regeneration of stem-cell mediated” This does not really make sense – “mediated by stem cells”?

Line 13 I would suggest using ‘medial regions’ instead of ‘middle’ here

Line 16 “missing epidermis”

Line 19 suggest “medial tissue regeneration”

Introduction

It would be useful to state that, prior to this study, the function of CK1a in planaria and in stem cells was not yet defined, to define the rationale of the study more clearly (as is given in line 308 in the Discussion).

Methods

Lines 85 and 89: Give the primer names for cDNA amplification and dsRNA generation respectively. This section should also include reference to slit dsRNA generation (as referenced in table S1).

Line 86 Give the source of PMD-19- vector, and the source of the cDNA (intact planaria?)

Line 90/91 Give the amount (ng or ug) and volume of dsRNA / water injected. Give the diluent that the dsRNA was dissolved in. Give the injection site. Was the injection site varied day to day, or the same? Was the injection once a day?

Line 115 Clarify the nature of the 3 samples: 3 planaria or 3 pools of planaria?

Line 121 Define SD as standard deviation on first use

Line 122 Explanation / language needs clarification: was the Student’s t-test or ANOVA used?

Results

Line 135 “facilely” - is this intended to mean superficial?

Fig 1 Was a positive or negative control RNAi used?

Fig 1B More information is needed. The number of planaria per condition is not given. Are the planaria pooled into WT and RNAi? What are the replicates? What stage planaria was this analysed at, i.e. after 7 days RNAi, before amputation or after? The SD for WT is very small, I would expect a larger SD for a technique like qPCR. There is no SD bar presented for the RNAi group, which needs to be added. It would be most useful to present the data points in each group in a box-and-whisker plot or a plot as in Fig 3D:a,b,c.

Fig 1D Clarify in the figure / legend that the control is amputated (i.e. not un-injured).

Line 159 It would be useful to comment in the authors’ response if the extended white line might be a malformation or extension of the pharynx in RNAi animals?

Line 173 This line is not clear in its meaning, please clarify.

Line 178 Suggest replacing ‘blurry’ with ‘more diffuse staining of the cephalic ganglion areas’ or similar.

Line 225 “where may arise enation” – I would suggest an alternative word such as outgrowth, as enation is not commonly used (as far as I am aware).

Fig 3A Include “ck1a RNAi” label in the figure.

Fig 3D qPCR: As above, it would be helpful to give the N number of samples here, and show the data points if possible.

Line 255 include “(uninjured)” or “(without amputation)” after “intact”, to clarify this point.

Line 276 use “regenerating heads” instead of “palingenetic”.

Fig 4E / line 294 It would be useful to show or state that the remaining 9/22 slit/ck1a RNAi heads showed one central eye (the slit phenotype).

Supplementary Figs

Fig S1 – A and B labels are not shown on the figure.

Table S1 – missing the table title/legend.

Round 2

Reviewer 1 Report

The authors have addressed the main concerns. 

The text should be revised for English stile and typographical errors.
